# Racial Disparities Persist Beyond Data Representation in Medical Imaging — even Predictive Uncertainty Fails to Capture them

**Tareen Dawood** [1]  iD                                                    TARDA@DTU.DK
**Gloria Stucchi** [1]                                         S232437@STUDENT.DTU.DK
**Aasa Feragen** [1]                                                          AFHAR@DTU.DK
[1] *Department of Applied Mathematics and Computer Science, Technical University of Denmark*

**Editors:** Accepted for publication at MIDL 2025

## Abstract

Balanced training sets are often promoted to mitigate racial performance disparities of Deep Learning (DL) models in medical imaging. However, **our preliminary findings on two medical imaging datasets show that while racial training set representation affects model performance, there is more at play, as large racial disparities remain regardless of training set composition.** Moreover, predictive uncertainty is shown to be entirely insensitive to these performance disparities, raising a series of open challenges for safe and fair image-guided diagnostics.

**Keywords:** Algorithmic Fairness, Bias, Uncertainty, Disparities, Representation

## 1. Introduction

Reliability of machine learning tools is an active area of research (Puyol-Antón et al., 2021; Hussain et al., 2022; Ricci Lara et al., 2022; Jiménez-Sánchez et al., 2023; Ferrante and Echeveste; Lekadir et al., 2025). However, inherent bias in training data is often overlooked, and classification outcomes tend to reinforce biases, replicating current and previous socioeconomic inequalities and produce errors that correlate with demographic variables or even potentially hidden attributes not explicitly available in collected data (Ferrara, 2023; Larrazabal et al., 2020). These systems could produce inadequate outcomes if they are deployed in real-world settings. Mitigation strategies have been proposed for unfair classifiers (Zong et al., 2022). However, a lack of understanding of the deeper causes of bias will limit performance and could perpetuate or even introduce new unwanted bias (Petersen et al., 2023). While Metha *et. al* (Mehta et al., 2024) evaluate fairness and uncertainty for subgroup performance, they vary dataset composition, a key aspect of our approach, for uncovering additional biases. Thus, this paper investigates how racial composition in training data impacts performance and uncertainty utilising two medical imaging datasets, assessing whether uncertainty and accuracy can reveal any disparities.

## 2. Methods and Experimental Design

We monitor racial performance disparities using a standard ResNet backbone across two different datasets:

- **RETINAL Dataset**: Consists of 2D retinal nerve fiber layer (RNFL) thickness images (200 × 200 pixels) curated with equal representations of Asian, White, and Black subjects across training (2,100), validation (300), and test (900) groups (Luo et al., 2024). The predictive task was a binary Glaucoma diagnosis classification.

- **PASSION Skin Imaging Dataset**: An unbalanced dermatological dataset curated to represent racial groups with darker skin tones (Gottfrois et al., 2024), containing 4,901 images (224 × 224 pixels) from 1,653 patients across five Fitzpatrick skin type (FST) phototypes. The predictive task was a multiclass classification to diagnose eczema, fungal infections, scabies, or other skin diseases.

**Training Data Composition:** Training subsets with varied racial or skin tone proportions (Larrazabal et al., 2020) were created using the smallest subgroup size (700 RETINAL, 1081 PASSION). Models were retrained independently with fixed settings; 10 seeds were used for RETINAL, and 5 for PASSION to isolate group-specific training effects.

A series of experiments to examined performance for different racial compositions in the training data, using similar hyperparameters as the original papers: Models were trained for 50 and 80 epochs with learning rates of $5 \times 10^{-5}$ and $1 \times 10^{-4}$ and batch sizes of 8 and 64 for the RETINAL and PASSION datasets, respectively. The representation of Asian, White, and Black subjects varied for RETINAL, while for PASSION, due to limited samples, we only varied the training representation of the lightest skin tone (FST 3). The configurations of the training sets were 20%, 40%, 60%, 80%, and 100% for each group, whilst the remaining samples were distributed equally across the race groups or skin tones.

**Performance Disparities:** To assess bias due to racial representation during training, we measure groupwise performance using the Area Under the Receiver Operating Characteristic Curve (AUC; RETINAL) and balanced accuracy (BACC; PASSION). Next, we check whether predictive (model) uncertainty can flag potential bias. To capture the uncertainty, we employed the popular Monte Carlo (MC) dropout ($p=0.3$) and the gold standard Ensemble methods (5 models) (Gal and Ghahramani, 2016; Rahaman et al., 2021). As measures of model uncertainties, these should flag performance loss due to samples being out of distribution, as one might expect for groups that are underrepresented during training.

## 3. Discussion and conclusion

Our **first** main finding, shown in Figures 1, is that while dataset composition does affect performance, this effect is relatively small, especially on the RETINAL dataset. This does not, however, mean that the models are fair; we see large performance disparities between races and skin color that persist regardless of training set composition. In both datasets, the black or dark skinned population is at a disadvantage. Our **second** main finding is that model uncertainty does not flag these performance disparities and is, therefore, not useful for flagging performance drops on racial subgroups. These findings support our main conclusion: **Uncertainty methods do not always reduce or explain racial performance disparities. Therefore, we hypothesize that there are further underlying causes for the differences** (Drukker et al., 2023). A deeper understanding of these underlying causes is, therefore, an important open problem.

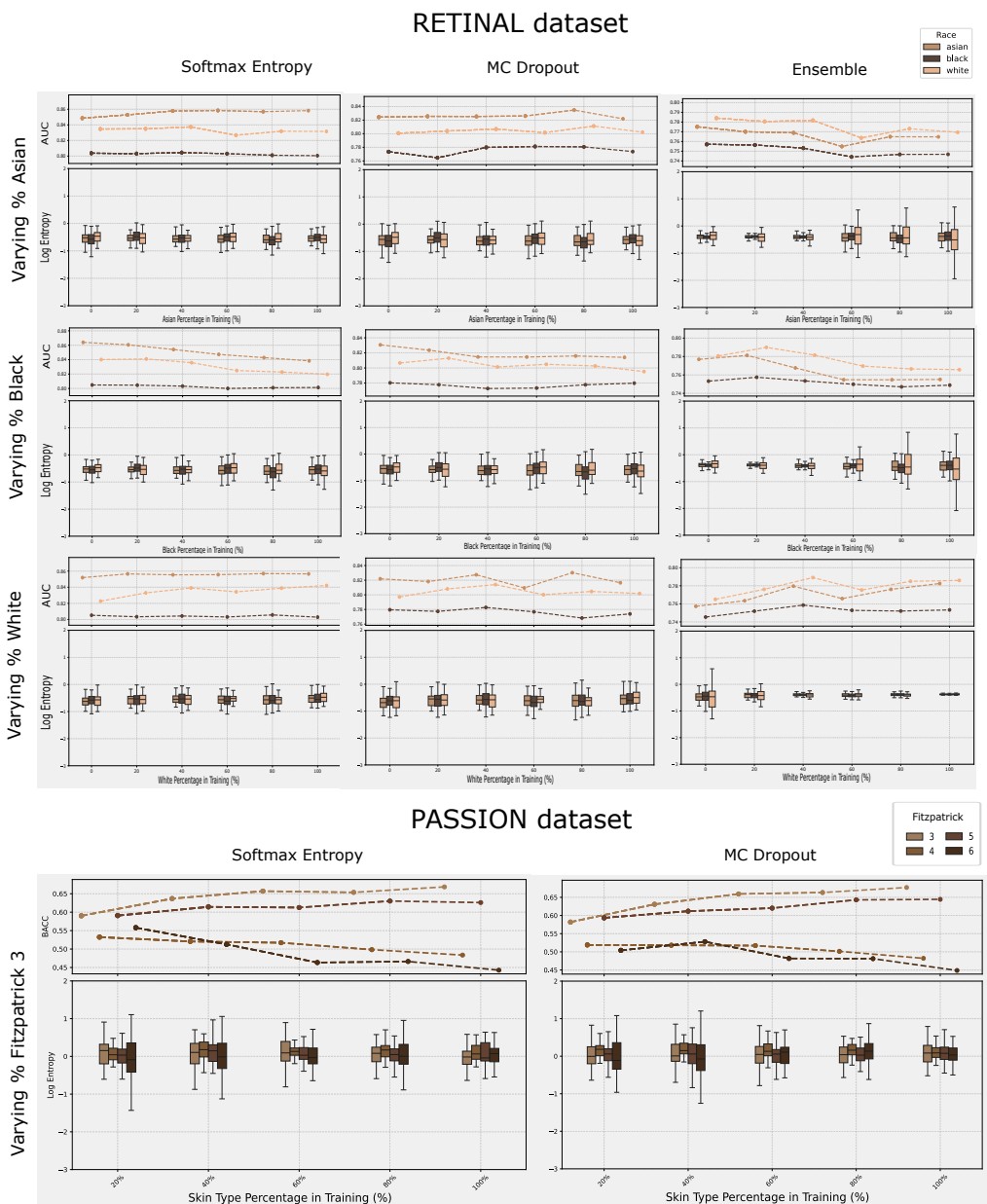

Figure 1: TOP: Retinal AUC trends vs. predictive uncertainty (log entropy) across columns; Asian (top), Black (middle), and White (bottom) demographics for models in each row; Base (left), MC Dropout (middle), and Ensemble (right) models trained on RETINAL datasets. BOTTOM: Comparison on PASSION dataset for Baseline (left) and MC Dropout (right) when trained on Fitzpatrick 3 but using BACC, with Ensemble and alternate methods left for future experiments.

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

## Appendix A. Additional Analysis

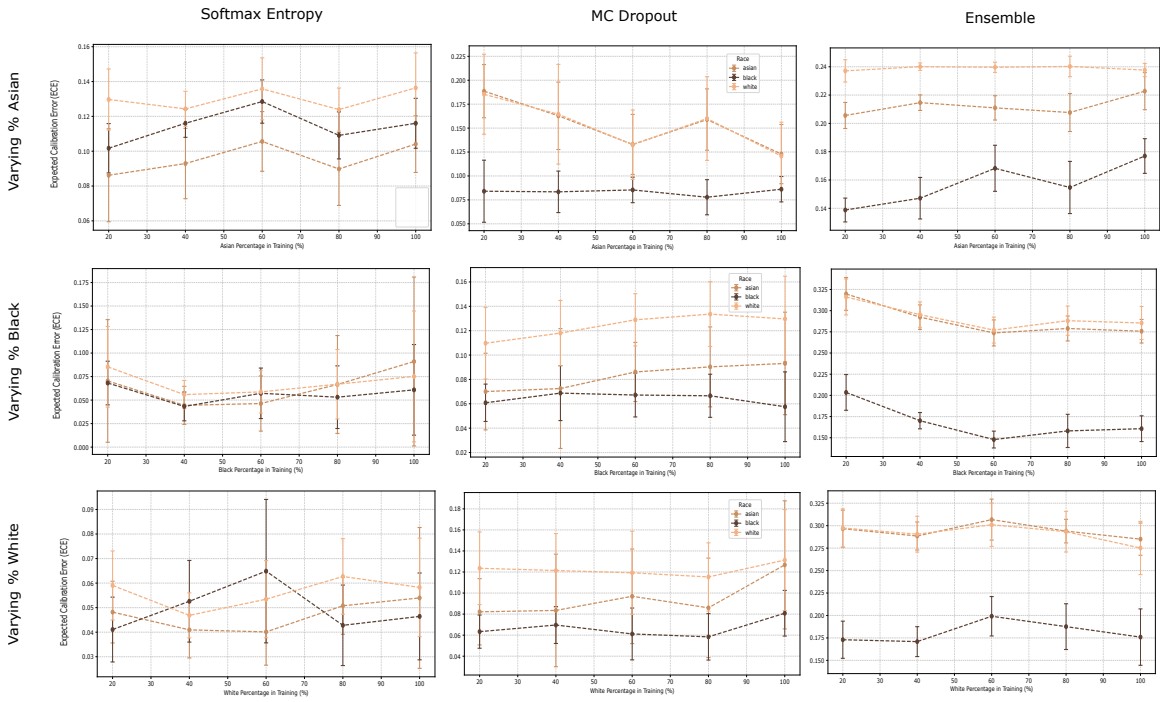

Figure 2: Reported Expected Calibration Error (ECE) for each subgroup (row) and each model (column) across the RETINAL dataset.

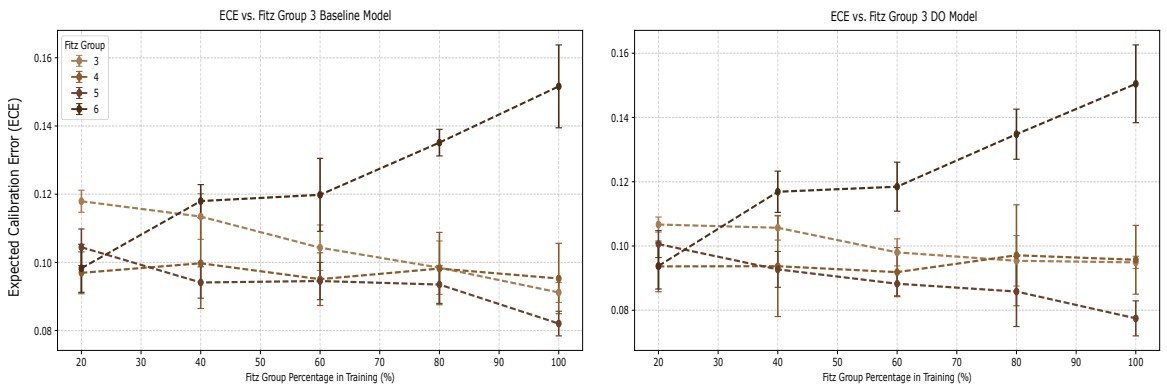

Figure 3: Reported ECE when training on the FST3 skin tone for the PASSION dataset.

