# OpenReview forum: "Racial Disparities Persist Beyond Data Representation in Medical Imaging — even Predictive Uncertainty Fails to Capture them"
_MIDL.io/2025/Short_Papers — MIDL 2025 - Short Papers_

### Official Review · Reviewer_PZGD · 2025-04-19

**Rating:** 4
**Confidence:** 4

**Summary:**

The paper proposes to analyze the fairness effect by varying the dataset size on two publicly available datasets. In addition to standard performance evaluation, uncertainty evaluation is also performed. Results indicate that while it is possible that standard evaluation metrics might be fair across demographics, it could affect the uncertainty associated with models.

**Strengths:**

* Clear goal and interesting insights regarding fairness and uncertainty.
* Experiments on two different datasets for a short paper are commendable.
* Well-presented results.

**Weaknesses:**

* **Missing reference**: Paper is missing a key reference that also analysed fairness from an uncertainty perspective [1]. I can see that, unlike this work, in that paper, the authors didn't vary the size of the training dataset; the performed analysis is quite similar to the current paper. It would be good to discuss this paper in the literature review section.
* **Dataset stratification**: How dataset stratification was done is unclear. For example, in the Retinal dataset, I can see that the authors varied the dataset percentage for each subgroup separately. But what was the original number of images per racial group? Also, when authors vary the size of one particular racial group, is the size of the other group kept similar? For the PASSOIN dataset, are all racial groups varied simultaneously or only one particular group?
* **Uncertainty metric**: I like the currently employed evaluation metric, but reporting something like calibration for each subgroup might be a good idea. For example, it could be the case that when accuracy is lower for a particular subgroup, their calibration might also be low. Another option could be using the rejection curve employed in [1,2].

[1] Mehta, R., Shui, C. and Arbel, T., 2024, January. Evaluating the fairness of deep learning uncertainty estimates in medical image analysis. In Medical Imaging with Deep Learning (pp. 1453-1492). PMLR.

[2] Kahl, K.C., Lüth, C.T., Zenk, M., Maier-Hein, K. and Jaeger, P.F., 2024. Values: A framework for systematic validation of uncertainty estimation in semantic segmentation. arXiv preprint arXiv:2401.08501.

---

### Decision · Program_Chairs · 2025-05-01

Accept